# Mask or no mask for COVID-19: A public health and market study

**Tom Li[1], Yan Liu[2], Man Li[1], Xiaoning Qian[3], Susie Y. Dai[1] \***

**1** Department of Plant Pathology and Microbiology, Texas A&M University, College Station, TX, United States of America, **2** Department of Marketing, Texas A&M University, College Station, TX, United States of America, **3** Department of Electrical and Computer Engineering, Texas A&M University, College Station, TX, United States of America

\* sydai@tamu.edu

**Data Availability Statement:** All relevant data are available at https://github.com/Environmentalpublichealth/covid-data-model.

**Funding:** The author(s) received no specific funding for this work.

## Abstract

Efficient strategies to contain the coronavirus disease 2019 (COVID-19) pandemic are peremptory to relieve the negatively impacted public health and global economy, with the full scope yet to unfold. In the absence of highly effective drugs, vaccines, and abundant medical resources, many measures are used to manage the infection rate and avoid exhausting limited hospital resources. Wearing masks is among the non-pharmaceutical intervention (NPI) measures that could be effectively implemented at a minimum cost and without dramatically disrupting social practices. The mask-wearing guidelines vary significantly across countries. Regardless of the debates in the medical community and the global mask production shortage, more countries and regions are moving forward with recommendations or mandates to wear masks in public. Our study combines mathematical modeling and existing scientific evidence to evaluate the potential impact of the utilization of normal medical masks in public to combat the COVID-19 pandemic. We consider three key factors that contribute to the effectiveness of wearing a quality mask in reducing the transmission risk, including the mask aerosol reduction rate, mask population coverage, and mask availability. We first simulate the impact of these three factors on the virus reproduction number and infection attack rate in a general population. Using the intervened viral transmission route by wearing a mask, we further model the impact of mask-wearing on the epidemic curve with increasing mask awareness and availability. Our study indicates that wearing a face mask can be effectively combined with social distancing to flatten the epidemic curve. Wearing a mask presents a rational way to implement as an NPI to combat COVID-19. We recognize our study provides a projection based only on currently available data and estimates potential probabilities. As such, our model warrants further validation studies.

## Introduction

During the COVID-19 pandemic that has significantly disrupted the global health system and economy, non-pharmaceutical interventions (NPIs) with potential public health benefits and little social and economic burdens should be promptly evaluated. Two Asian countries (China

**Competing interests:** The authors have declared that no competing interests exist.

and South Korea) have widely recommended wearing a mask to manage the spread of the severe acute respiratory syndrome coronavirus 2 (SARS-CoV-2) that leads to COVID-19 [1]. This practice has been widely debated in other countries, as some previous experimental studies on other respiratory diseases such as influenza H1NI suggested the limited effectiveness of using face masks to prevent infection [2]. However, risk assessment studies using population transmission models suggested that the population-wide use of face masks could delay an influenza pandemic [3]. Furthermore, effects studied in closed settings (aircraft or households) provided preliminary evidence that masks can contribute to infection prevention [4, 5].

Different from influenza virus, SARS-CoV-2 is a recently discovered virus in the coronavirus family. SARS-CoV-2 is more closely related to the other two coronaviruses that led to two recent outbreaks, SARS-CoV-1 that caused the severe acute respiratory syndrome (SARS) outbreak in 2002 and MERS-CoV (Middle East Respiratory Syndrome Coronavirus) that caused the MERS outbreak in 2012 [6]. As a novel virus, the SARS-CoV-2 transmission features have yet to be fully characterized. There is strong increasing evidence that SARS-CoV-1 and SARS-CoV-2 could be airborne [7–9]. As a result, the effect of using masks to combat SARS-CoV-2 is under evaluation [10, 11].

In recognition of the global personal protection equipment (PPE) shortage, we explore the impact of medical face masks (loose-fitting surgical masks) on controlling virus spread in the current pandemic. We investigate three factors that might influence the effectiveness of mask use and the COVID-19 transmission rate, including the mask aerosol reduction rate, mask availability, and mask population coverage. We then evaluate the impact of wearing face masks on flattening the epidemic curve. We parameterize the face mask effects based on available scientific evidence and simulate the impact throughout the pandemic. Our findings are consistent with the WHO's advice on the use of masks in the context of COVID-19 [12].

## Model and analysis

We use the following equation to predict the basic reproduction number $R_0$ of the COVID-19 pandemic:

$$R_0 = b \, \kappa \, D,$$

where $b$ is the transmission risk per contact, $\kappa$ is the contact rate (numbers per time period) between an infected and susceptible individuals in the population, and $D$ is the duration of infectivity of an infected individual measured in the same time unit used for $\kappa$ [13].

We then estimate the effect of mask-wearing on the $R_0$ of the pandemic. We presume wearing masks decreases the risk of contracting the virus depending on the population mask effectiveness (= $M_{eff}$). In terms of viral infectious dose, the exposure reduction by wearing a mask can proportionally decrease infection risk [3, 14] and, consequently, the virus transmission in the general population. It is reasonable to estimate that SARS-CoV-2 transmission may only involve small doses due to its high infectivity and relatively high reproduction number [15, 16].

Therefore, if masks are properly used in the population, then the probability of transmission per contact $b$ will be reduced by the fraction of $M_{eff}$. $R_0$ will be reduced by this same fraction. We define the new decreased reproduction number as $R_{int}$ based on mask intervention. Thus,

$$R_{int} = (1 - M_{eff}) \, b \, \kappa \, D$$

The effectiveness of a mask ($M_{eff}$) in reducing $R_0$ depends on the following three key factors. First, the "mask availability" within a population (= $M_{ava}$). Mask availability indicates the

proportion of a population that has access to a mask. Face mask shortage is a considerable problem faced by many countries during the ongoing COVID-19 pandemic [17, 18]. Therefore, we do not expect each individual to have access to a mask during an outbreak. Second, "mask coverage" (= $M_{cov}$) is conditional on the mask availability to the population ($M_{ava} = 1$). "Mask coverage" measures the proportion of appropriate mask use within a population conditional on mask availability. The third factor is the aerosol reduction of mask $M_{red}$ conditional on proper mask usage ($M_{cov} = 1$). The aerosol reduction rate $M_{red}$ captures the mask's virus filtering efficiency, that is, the proportion of pathogenic organisms filtered by the mask. Given that $M_{cov}$ and $M_{red}$ capture the conditional probability of mask coverage and aerosol reduction respectively, the joint probability of (1) masks available to the population, (2) properly used by the population, and (3) being effective in filtering the virus is simply the production of these three factors. Therefore, the $M_{eff}$ equation is expressed as

$$M_{eff} = M_{red} * M_{cov} * M_{ava}$$

We then rewrite the equation for the reproduction number:

$$R_{int} = (1 - M_{red} * M_{cov} * M_{ava}) \, R_0.$$

We assume a random mixing model and use the following equation to estimate the effects of mask use on the infection attack rate $a$ [3]:

$$a = 1 - e^{-aR_{int}}$$

During the first wave of a pandemic, the entire population is susceptible. The attack rate $a$ is the infected population proportion after the first wave. As the number of infectious contacts per infection is $R_{int}$, the total number of infectious contacts during the wave per person is $aR_{int}$. Assuming random mixing, the probability that an individual is not contacted by any infectious person is $e^{-aR_{int}}$. Therefore, the probability that an individual is contacted by at least one infectious person is $1 - e^{-aR_{int}}$. In the beginning, each individual is equally susceptible. Thus, $1 - e^{-aR_{int}}$ also expresses the probability that an individual is infected, which is equal to the infection attack rate $a$.

We use a compartmental epidemiological model developed by Hill et al. [19], a classic SEIR model, to simulate disease outbreaks in both research and practical settings. The model simulates individuals progressing through several compartments or states: susceptible (S), the infection's start state; exposed (E), where they have been infected but are not yet symptomatic or contagious; infected (I), where the individual shows symptoms ranging from mild to severe; recovered (R), for individuals who have already had the disease and are assumed to be immune; and deceased, for those who do not survive the disease. The model is based on the process that 1) susceptible (S) individuals may become exposed (E) to the virus, 2) infected (I) at varying levels of disease severity, and 3) the infected individuals either recover (R) or die (removed). Notably, some evidence suggests that COVID-19 infection may not create long-term immunity in some individuals [20]; however, our model assumes that recovered patients cannot regain the infection for the duration of the model.

We implement several changes and simplifications to the model to focus on the effects of masks on infections. First, we do not consider the data for individual states but model the US population as a whole. We do not model the effects on hospitalizations or deaths, as those are assumed to be a flat percentage of infections. We modify the various intervention scenarios corresponding to our different mask scenarios and adjust the simulation's end date. The implemented simulation can be accessed in the public domain [21]. The model assumptions and constraints can be modified in the code if needed.

Table 1. Key virus transmission parameters.

| Disease | $R_0$ | Incubation period T (days) | Transmission pathways |
|---------|-------|----------------------------|----------------------|
| **COVID-19** | 2.2 [31] | 3.9 [34] | Direct contact, airborne (under study), droplets |
| | 2.28 [15] | 5.1 (4.1–5.8) [35] | |
| | 1.05–2.35 [32] | 5.2 (SD: 3.7) [32] | |
| | 2.76–3.25 [33] | | |
| | 6.47 (5.71–7.23) [16] | | |
| | 14.8 [30] | | |

## Virus features

COVID-19 transmission mechanisms and dynamics are still under investigation [8, 22]. It has been recognized that bio-aerosols generated directly by patients' exhalation could spread SARS-CoV-2. The potential transmission routes could include air droplets and aerosols as viral RNA has been detected in both matrices. Although the presence of viral RNA does not necessary warrant active virus presence and thus transmission, precautionary measures are suggested in those earlier COIVD-19 outbreak areas [23, 24].

Among existing virus transmission feature studies, we compare COIVD-19 to flu, SRAS, MERS, and other highly contagious infectious diseases such as measles. Literature reviews reveal more available data on influenza [25–27]. In one influenza study, influenza virus RNA was detected in the exhaled breath of 33% of influenza patients while most people exhale more than 500 particles per liter of air [28]. Particularly, droplet particles are larger than aerosols when regular exhalation mainly results in aerosol production [29] ($> 99\%$ of exhaled particles $< 5 \mu$m). The recent estimated $R_0$ value for COVID-19 is approximately 2.3, which is consistent among several studies (Table 1). The highest estimated $R_0$ value is 14.8 [30] based on a cruise ship study. However, the other group reports a lower $R_0$ of 2.28 in the early stage from the same cruise ship [15].

## Asymptomatic and pre-symptomatic individuals and their impact on transmission

There are documented cases remaining asymptomatic throughout the duration of laboratory and clinical monitoring [34, 36–39]. In many cases, a significant portion developed some symptoms at a later stage and thus are "pre-symptomatic" [40–45]. Studies also suggested that asymptomatic patients could spread the virus as their viral loads have no significant differences compared to those of symptomatic patients [46, 47]. As such, pre-symptomatic transmission was estimated to have a shorter serial interval of COVID-19 (4.0 to 4.6 days) than the mean incubation period (five days) [12]. As a result, many secondary transmissions could have happened before the symptomatic cases were detected and isolated [48, 49]. Notably, Taipiwa et al. reported the pre-symptomatic transmission of 48% (95% CI 32–67%) for the Singapore outbreak and 62% (95% CI 50–76%) for the Tianjin outbreak in China [50].

## Mask features

Masks can play at least two roles in viral transmission prevention in the general population. First, masks can impact turbulent gas cloud formation and respiratory pathogen emission [51]. Research demonstrates that masks can either block the rapid turbulent jets generated by coughing or redirect the jets in much less harmful ways for airborne infection control [52]. Second, the mask material can filter viral particles such as aerosols or droplets [53]. Additionally, for asymptomatic infected individuals, wearing a mask can potentially reduce the risk of

infecting other people when the exact individual wears a mask to protect him or herself. We classify masks into three categories in our study: 1) certified masks, which refers to medical masks that meet government certification standards (that is, in the US, the Centers for Disease Control and Prevention (CDC) National Institute for Occupational Safety and Health (NOISH) certifies medical masks such as N-95 respirators); 2) medical masks that are not certified but subject to Food and Drug Administration (FDA) jurisdiction as a regulated medical device (that is, loose fitting disposable medical masks); and 3) homemade masks whose quality cannot be guaranteed. For medical masks, the European Medicines Agency (EMA) has established guidelines for effective virus reduction combined with reduction factors [54]. Typically, certified and medical masks can effectively reduce influenza virus loads [55]. Leung et al. described the effectiveness of surgical face masks (with ear loops, cat. no. 62356, Kimberly-Clark) could prevent the transmission of human coronaviruses and influenza viruses from symptomatic individuals. Based on the guidelines and available data on certified and non-certified medical masks, we recognize that the mask material virus reduction potential is not necessarily equivalent to the mask viral reduction rate ($M_{red}$) and we assume that the general mask usage in public has a reduced reduction factor without any fit tests, training, or instructions. Previous studies compared homemade cloth masks and commercial medical masks, which suggested reduced protection from particle penetration by cloth masks [56–58] and bare protection by handkerchiefs [57]. However, cough pressure can be significantly reduced by wearing any type of mask [59]. As such, we estimate less than 1–2 log 10 reduction factors for normal mask wearing in public. The log reduction factor translates into less than 90% virus removal effectiveness. We assume $M_{red}$, the base aerosol reduction percentage of face masks (commercial medical products) in a public setting, to be approximately 60% [60] and estimate the range from 40% to 75%, assuming the best reduction rate is 99% for a NOISH-certified N-95 type respirator [53, 57, 58, 61].

The percentage of people wearing a mask during a pandemic depends on several factors. First, culture plays a very important role in determining mask coverage around the world [62]. In East Asia, wearing a mask is common and has long been culturally acceptable [62]. People wear masks for many different reasons, such as pollution, allergies, and winter protection, not just when they are sick. According to a recent Mintel report, 63% of Japanese wore face masks in public during the spread of COVID-19 [63]. However, in North America and Europe, public health officials have discouraged healthy people from wearing masks [62]. Previous studies across five countries suggested a significant gap between willingness (71%) and real action (8%) to wear a mask in the US [64]. The awareness of wearing a mask during the COVID-19 pandemic recently became more popular in the US, and the percentage of Americans wearing masks increased to approximately 12% by the end of March 2020 [65]. Therefore, we expect $M_{cov}$ to be higher in East Asian countries and lower in North America and Europe. Second, the pandemic's severity could potentially change the mask coverage dynamics in a short period. Based on an online (February 11 to 13, 2020) survey in South Korea, 79% of the participants started wearing masks compared to only 19% who wore masks prior to the outbreak. Third, public health advocacy and government policies or recommendations could have a significant impact on mask coverage. To simulate the mask coverage impact on $R_0$, we assume $M_{cov}$ to be in a range of 8% to 100%.

Most countries were not prepared for the COVID-19 outbreak and are universally short of PPE supplies. Abramovich et al. utilized a computer simulation to model the benefits of stockpiling PPE based on disease profile variables. The simulation variables provided a wide range that covered the current COVID-19 outbreak, whose disease parameters fall into the higher end of the range. The authors suggested diminishing patient care benefits of stockpiling on the high side of the range [66]. However, the study pointed out the importance of having modest

stockpiles of critical resources. Carias et al. estimated in a hypothetical influenza outbreak that 1.7 to 3.5 billion respirators would be needed in the base case scenario, 2.6 to 4.3 billion in the intermediate demand scenario, and up to 7.3 billion in the maximum demand scenario for an outbreak with 20% to 30% of the population infected. Among all of the scenarios, between 0.1 and 0.4 billion surgical masks would be needed for patients [67]. Moreover, the production of N95 respirators and other surgical masks has increased since the COVID-19 outbreak. As of February 3, 2020, it was estimated that China was producing approximately 14.8 million medical masks daily, a production capacity utilization rate of nearly 67% [68]. The future trend will follow the market needs, government agency public health policies, and supportive programs [69]. Given the limited data and vast uncertainty of future mask production, we estimate $M_{ava}$ varies significantly across countries and regions. In countries with a large mask production capacity such as China [70], $M_{ava}$ is on the higher end, approximately 90%. However, countries that rely heavily on importing face masks, such as Switzerland, are more likely to face shortages because of the surging global demand and disrupted global supply chain [18]. As a result, $M_{ava}$ for these countries could be on the lower end, approximately 30%. Moreover, we project that mask availability will increase as the pandemic peaks and production continues to rise.

We used the following table for the $R_{int}$ simulation of the mask features. We set $M_{red}$ at 57.5% and considered two parameters for $M_{cov}$ and $M_{ava}$ individually for the simulation. We set the baseline scenario of 8% for $M_{cov}$ [64] and 100% as the best scenario. Given the shortages in the PPE market, we used 5% as the baseline scenario and 100% as the best scenario. We simulated seven scenarios (Table 2) and discuss the details.

## Results and discussion

### Effects of mask-wearing on reproduction number and infection attack rate

Based on the reported studies, we set $R_0$ at 2.3 to evaluate the mask impact. As previously mentioned, we exclude homemade face masks from this evaluation as the mask material and quality cannot be guaranteed. To show how the reproduction number $R_{int}$ and infection attack rate $a$ are impacted by mask-wearing, we plot the change in $R_{int}$ and $a$ with mask availability $M_{ava}$ under seven scenarios. We report the values of $M_{red}$ and $M_{cov}$ for these seven scenarios (S1 to 7) in Table 2. Fig 1 shows that $R_{int}$ decreases with mask availability in all of the scenarios. Specifically, in scenarios 2 and 5, when everyone is willing to wear a mask ($M_{cov} = 100\%$), $R_{int}$ is among the lowest (that is, $R_{int}2$ and $R_{int}5$). It can be less than 1 when mask availability is close to 100%. Moreover, even a moderate level of mask coverage ($M_{cov} = 54\%$, scenarios 3 and 7) can help substantially reduce $R_{int}$ (i.e., $R_{int}3$ and $R_{int}7$) compared with low mask coverage ($M_{cov} = 8\%$, $R_{int}1$, $R_{int}4$, and $R_{int}6$). We observe a similar pattern in the infection attack rate $a$

**Table 2. Parameters for reproduction number, infection attack rate, and infected cases in seven scenarios (S1 to S7).**

|  | $M_{red}$ | $M_{cov}$ | $M_{ava}$ |
|---|---|---|---|
| **Median (range)** | 57.5% (40%-75%) | 54% (8%-100%) | 52.5% (5%-100%) |
| **S1** | 57.5% | 8% | 5% |
| **S2** | 57.5% | 100% | 100% |
| **S3** | 57.5% | 54% | 52.5% |
| **S4** | 40% | 8% | 5% |
| **S5** | 75% | 100% | 100% |
| **S6** | 75% | 8% | 52.5% |
| **S7** | 75% | 54% | 5% |

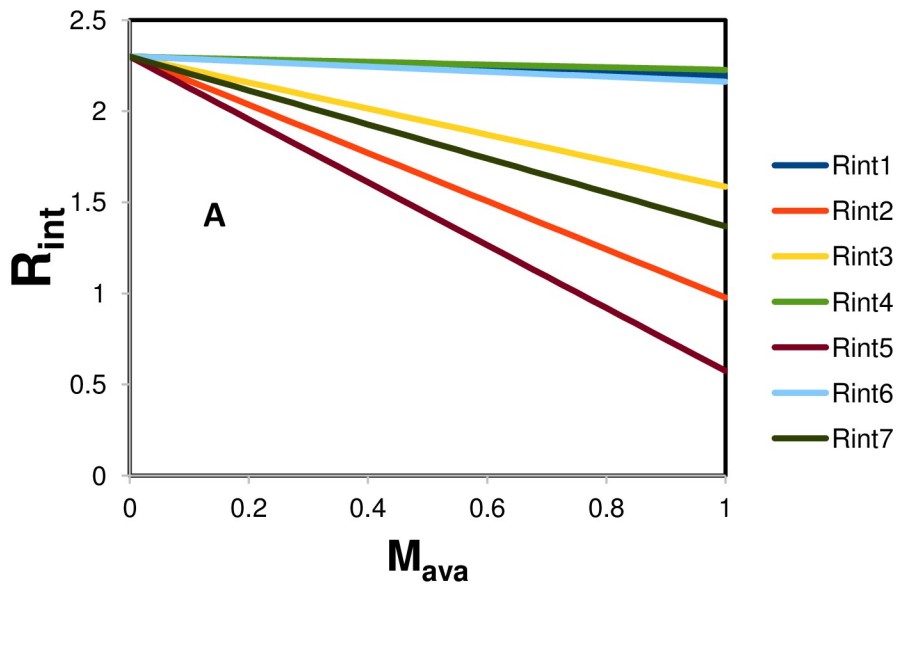

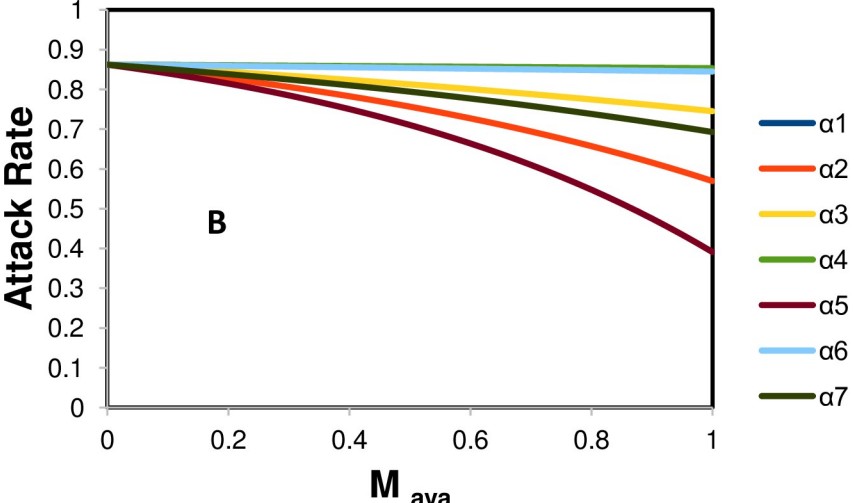

**Fig 1. $R_{int}$ and attack rate dependence on mask availability.** The $R_{int}$ and attack rate a values are simulated based seven scenarios in Table 2. $R_{int}$ 1 is calculated based on scenarios 1. The same annotation principle applies to all other $R_{int}$ calculations.

graph (Fig 1). These results indicate the significance of mask-wearing, demonstrating considerable promise to contain the pandemic.

## Transmission model

Based on the SEIR model [19], we further evaluate if wearing masks is an additional NPI and how this measure in combination with social distancing can help contain the pandemic and have a less catastrophic impact on the hospital system. Specifically, the SEIR model simulates the infected cases over time given an estimated initial $R_0$, modeling interval, recovery period, non-contagious incubation period, contagious period, and serial interval. We assume that $R_0$ is 2.3 without intervention [15, 30, 36], with 20 initial cases, a four-day modeling interval

(disease doubling period) [15, 32, 35], a 16-day recovery period, a two-day non-contagious incubation period and two-day contagious period, an average time of four days between symptom onset in patient one and the symptom onset of another individual infected by patient one, and a four-day average hospital stay. We also model that the infectivity level changes when the patient is admitted to the hospital (considered isolated). Based on earlier actual COVID-19 data fittings, it is estimated that the infection rate is in a range of 0.65–0.8 [71]. In our modeling, we set the rate to be 0.6 before hospital admission. We set the rate to be 0.1 after the individual is admitted. These hypothetical numbers are used to evaluate interventions at $R_0$ and the total infected cases throughout a pandemic over time. We first obtain two control conditions when there is no intervention (the blue dashed curve labeled "all-infected 2.3" in Fig 2A–2C) and when there is a relaxed social distancing situation assuming that $R_0$ is 1.7 for the first three months (the solid red curve "all-infected s_d" in Fig 2A–2C). We find that the outbreak without any intervention ("all-infected 2.3") and relaxed social distancing ("all-infected s_d") will lead to more than 40 million and 20 to 30 million infected people, respectively. In late April, it was projected that the US may have reached the peak of the epidemic curve. On April 20, 2020, the US had 759,687 confirmed cases based on data reported from the Center for Systems Science and Engineering (CSSE) at John Hopkins University [72]. Across the US, public measures against COVID-19 vary among states and cities. However, many cites (including New York City, Los Angeles, San Francisco, and Chicago) issued shelter-in-place or stay-at-home orders. On June 14, 2020, the US had 2,081,296 confirmed cases when many states reopened, with some cities and states recording increasing rates of new coronavirus cases per day. Although our model utilizes random mixing and assumes that all hosts have identical rates of disease-causing contacts with a homogenous nature, we expect that individual behaviors and heterogeneity would profoundly impact the epidemic curve across the country [73]. As a result, our model outputs could be higher than the real cases at a given time point. However, our purpose is to model face masks' impact on the epidemic curve in a general population without considering other compounding factors. The SEIR model thus represents a good rationale to evaluate face mask impact.

To show the effect of mask-wearing on the epidemic curve, we plot the infected cases over time with the seven previously discussed scenarios (Table 2) and compare them with the two control conditions in Fig 2A–2C. Fig 2A shows the three scenarios' epidemic curves when the mask filtering quality is at an average level ($M_{red}$ = 57.5%) compared with the two control conditions. With intermediate values of $M_{cov}$ and $M_{ava}$ between 50% to 60% (S3), we observe that the epidemic curve can be significantly flattened with an estimate of between 10 to 20 million people becoming infected (the maroon solid curve "all-infected_S3" in Fig 2A). However, if only 8% of the population is willing to wear a mask and only 5% can obtain masks (S1), masks have little impact on viral transmission intervention in a general population (the yellow dotted curve "all-infected_S1" in Fig 2A). In scenario 2, when everyone wears a mask, the pandemic can be efficiently managed (the green solid curve "all-infected_S2" in Fig 2A).

We plot the epidemic curve for S4 and S5 assuming that the three mask effectiveness factors are either the highest (greatest intervention) or the lowest (least intervention) in Fig 2B. If 8% of the population is willing to wear a mask and only 5% can obtain a poor-quality mask, the situation does little to contain the outbreak (the yellow dotted curve "all-infected_S4" in Fig 2B). However, if 100% of the population is willing to wear a mask and can obtain the highest quality mask, the impact of mask-wearing on containing the outbreak is the most effective among all of the scenarios (the green solid curve "all-infected_S5" in Fig 2B).

We further consider the last two scenarios with the mixed combination of $M_{cov}$ and $M_{ava}$ when one is low and the other is at an average level. These two scenarios reflect the situations when people are willing to wear masks when the mask availability is low, such as in Japan and

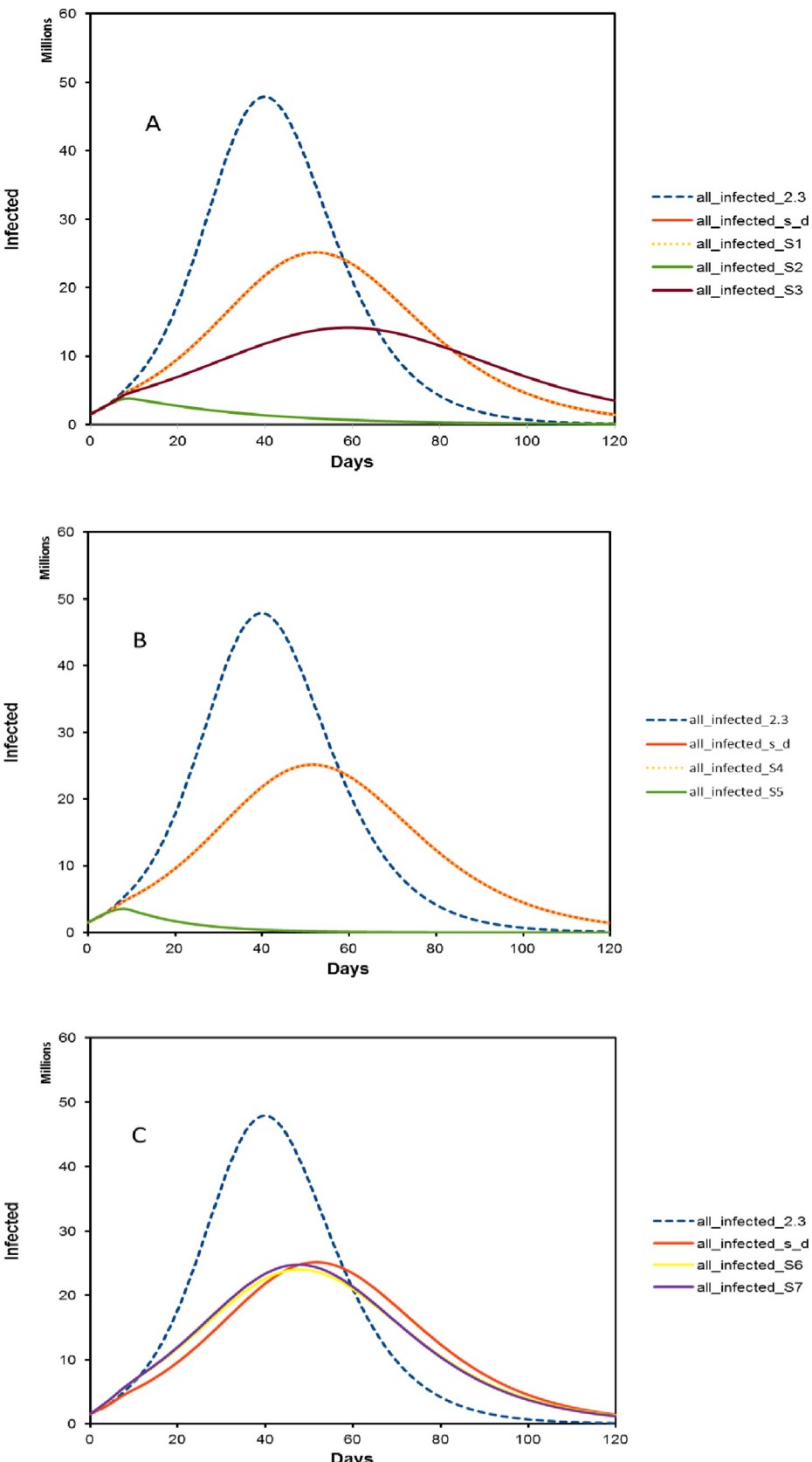

**Fig 2. Simulation on infected cases based on $R_{int}$.** All figures use two hypothetical controls. The blue dash line curve "all-infected 2.3" is simulated using $R_0$ value of 2.3. The red solid curve "all-infected s_d" is simulated using $R_0$ of 1.7 with relaxed social distancing, assuming a rough extrapolation of reducing about 50% of overall transmission risks in the general population. Fig 2A shows the applying social distancing and wearing a mask in three scenarios (S1,S2, and S3) when $M_{red}$ is 57.5%. Fig 2B shows the applying social distancing and wearing a mask in two scenarios (S4 and S5) for two extreme conditions. S4 is the scenario when $M_{red}$ = 40%, $M_{cov}$ = 8% and $M_{ava}$ = 5%.S5 is the scenario when $M_{red}$ = 75%, $M_{cov}$ = 100% and $M_{ava}$ = 100%. Fig 2C shows the applying social distancing and wearing a mask in two scenarios (S6 and S7) for two intermediate conditions, S6 with $M_{red}$ = 75%, $M_{cov}$ = 8%, $M_{ava}$ = 52.5%, and S7 with $M_{red}$ = 75%, $M_{cov}$ = 54%, $M_{ava}$ = 5%.

other Asian countries without mass mask production capacity, or vice versa. We find that, compared to the control conditions, if one of the $M_{cov}$ and $M_{ava}$ values is in the lower end range (less than 10%), mask-wearing has a minimum impact on containing the pandemic (the yellow solid curve "all-infected_S6" and purple solid curve "all-infected_S7" in Fig 2C).

In summary, we find that the simulated epidemic curve is sensitive to $M_{cov}$ and $M_{ava}$ in the general population (Fig 2). Mask coverage and availability play significant roles in the simulation, impacting the projected infected case numbers.

## Mask or no mask: Study significance and limitations

**Implications.** Our study presents a simulated quantitative analysis to evaluate the impact of wearing a mask on the reproduction number of viral infection and in turn the COVID-19 epidemic curve. We utilize the rather simple SEIR model to present the potential differences in the epidemic curve when using quality masks appropriately [3]. This study suggests that wearing a mask can potentially decrease the viral reproduction number in a general population. Wearing a mask in combination with social distancing and other measures is promising to replace the shelter-in-place orders and significantly reduce the COVID-19 burden on society.

We use the SEIR model developed by Hill et al. [19] and adopted by Henderson et al. [74]. Internationally, many other reported epidemic models are used to predict the outlook for COVID-19. Ferguson et al. modeled the NPI impact on mortality reduction and healthcare demands [75]. The study used a stochastic, spatially structured individual-based simulation to evaluate five NPIs individually or in combination. The five NPIs included case isolation, home quarantine, social distancing of seniors (older than 70 years old), social distancing of the entire population, and school closure. Koo et al. used FluTE, a model accounting for demography, host movement, and social contact rates in different social settings and assuming three $R_0$ values (1.5, 2.0, and 2.5).The FluTE model evaluates intervention strategies that include quarantine of individuals/households, quarantine plus immediate school closure for 2 weeks, quarantine plus immediate workplace distancing, and a combination of quarantine, immediate school closure, and workplace distancing (hereafter referred to as the combined intervention) [76]. Collectively, many of the current models estimate the $R_0$ value based on reports in earlier outbreak regions. As such, we expect regional variations and locality to significantly impact the $R_0$ value. However, very few studies have considered the potential utility of wearing a mask to decrease the respiratory virus reproduction number as a potential NPI.

We further plot confirmed cases over time for several countries or regions and compare the COVID-19 cases per million within the first 30 days (1 month) after the country or region reaches one case per million people. Two countries and two regions in Asia that had early confirmed cases were selected for comparison and used as the mask-wearing group. Five countries from EU were chosen as examples of the no-mask wearing group. Some Asian countries and regions such as Japan implemented wearing masks as an NPI to contain COVID-19 [1] since Japan reported its first case on January 16 [77, 78]. Thailand reported 25 confirmed COVID-19 cases on February 6 [79]. The Thailand Public Health Ministry strongly advocates wearing

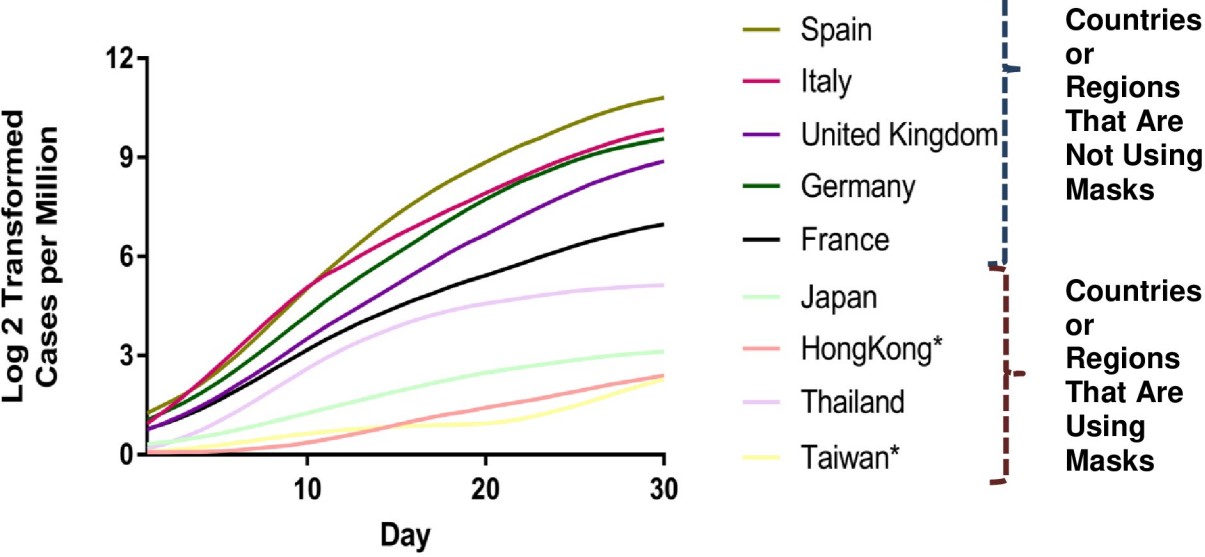

**Fig 3. Confirmed cases indifferent countries.**

masks and offers free masks to tourists [80]. On January 21, Taiwan reported its first imported COVID-19 case [81] and implemented multiple measures, including advocating wearing masks to mitigate the risk. As shown in Fig 3, five countries in the non-mask wearing group had a significantly higher increases in COVID-19 cases. A log transformed analysis is conducted to evaluate the growth rate of COVID-19 infections. ANOVA analysis is used to analyze the group effect. The p value is less than $2e^{-16}$ for the log 2 transformed case comparison, suggesting that the two groups are statistically significantly different. The Asian mask-wearing group clearly has lower growth rates of COVID-19 cases compared to the non-mask-wearing group. Thailand has the highest growth rate among the mask-wearing groups.

However, the differences in the growth rate of confirmed cases between these two groups cannot be fully attributed to mask-wearing, but arise from many factors, such as clinical tests performed, complicated social, economic, and cultural differences, and varied public health polices enforcement. The preliminary comparison only provides the rationale to further evaluate if wearing masks is effective for controlling the infectious disease outbreak.

**Model rationale and limitations.**   Our study has six major assumptions with preliminary evidence. First, the SARS-CoV-2 transmission pathway includes airborne [1, 8, 22–24] with an $R_0$ value exceeding that of influenza and SARS [16, 32, 33]. Second, a proportion of infected cases could be asymptomatic [36, 38, 46, 47]. Third, asymptomatic individuals could transfer the virus in the community [48, 49, 82]. Fourth, the public awareness of mask usage may increase as the pandemic spreads [64]. Fifth, the market could respond to consumer demand with a production increase and without supply chain problems [67]. Sixth, homemade cloth masks are not as effective as commercial medical masks [57, 60]. Thus, we project the best scenario as the pandemic spreads and by the end of the outbreak, the general population will be willing to wear a mask. Although wearing a mask is a low-cost and non-disruptive measure, wearing a mask has not been culturally widespread in many parts of the world [83]. Furthermore, for non-certified medical masks, individual errors could lead to ineffective mask fitting, which could reduce the benefit of wearing a mask [84].

There are limitations to those six assumptions. First, a recent study suggested that the virus was detected on the outer layer of COVID-19 patients' masks [85], which could invalidate the

utility of a patient wearing a mask. However, although the virus was detected, the mask could still provide prevention by reducing the virus load to an adjacent healthy individual. Further studies on asymptomatic individuals wearing masks could be more convincing to validate the effectiveness of wearing a mask to prevent pre-symptomatic transmission. Second, we assume random mixing and model the general population, but have not taken account children's role. Children are less likely to wear masks consistently to realize mask effectiveness. At the same time, the role of children in COVID-19 transmission is unclear [86]. Third, we assume the willing individual who has a mask can use the mask in the right manner and follow social distancing and other personal hygiene practices. There are discrepancies between willingness, perceived compliance, and real compliance [87]. Fourth, research shows that masks can be a source of contamination if not disposed of or replaced in timely manner [88].

## Conclusions

The COVID-19 pandemic has created a global crisis. Prevention such as vaccines is one of the most effective measures to mitigate such a catastrophic public health crisis. Prior to an available vaccine, NPIs such as wearing a mask can potentially reduce the virus transmission rate. Recent modeling studies suggest that timely and comprehensive NPIs are needed to prevent a secondary wave of COVID-19 [89]. However, timely implementations of NPI such as wearing a mask call for public awareness, a readily available market stockpile, and government advocacy and policies.

For respiratory diseases caused by a rival agent through aerosol and droplet transmission routes, wearing masks could be a reasonable NPI to reduce the virus transmission efficiency and secondary transmission [5, 89]. However, masks do not replace social distancing and other personal hygiene practices, such as hand washing. The effectiveness of any NPIs depends on compliance rates, contact rate reduction, the role of children, and asymptomatic cases in transmission. Our model analyzes the impact of wearing a mask that can efficiently filter viral aerosols during the COVID-19 pandemic in the general population. Our study suggests that wearing a quality mask in combination with other NPIs can support hospital resource management during a pandemic. In a free market, the risks of not having enough masks for the general population may negatively impact timely responses to the outbreak. Appropriate public health policies and government subsidies could be necessary to manage similar crises in the future. Future studies on how to improve adherence and compliance rates will better position society for potential future respiratory disease outbreaks when the viral agent has a similar clinical attack rate.

## Author Contributions

**Conceptualization:** Yan Liu, Susie Y. Dai.

**Data curation:** Tom Li, Man Li, Susie Y. Dai.

**Formal analysis:** Tom Li, Man Li, Xiaoning Qian, Susie Y. Dai.

**Methodology:** Tom Li, Yan Liu, Man Li, Xiaoning Qian, Susie Y. Dai.

**Software:** Tom Li, Xiaoning Qian.

**Supervision:** Yan Liu, Susie Y. Dai.

**Writing – original draft:** Tom Li, Yan Liu, Man Li, Xiaoning Qian, Susie Y. Dai.

**Writing – review & editing:** Yan Liu, Xiaoning Qian, Susie Y. Dai.

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
