## [Decision Letter · Decision Letter 0]

9 Jun 2020

PONE-D-20-11539

Mask or No Mask for COVID-19: a Public Health and Market Study

PLOS ONE

Dear Dr. Dai,

Thank you for submitting your manuscript to PLOS ONE. After careful consideration, we feel that it has merit but does not fully meet PLOS ONE’s publication criteria as it currently stands. Therefore, we invite you to submit a revised version of the manuscript that addresses the points raised during the review process.

We look forward to receiving your revised manuscript.

Kind regards,

Kednapa Thavorn, PhD

Academic Editor

PLOS ONE

Journal Requirements:

2. Please ensure that all statements are supported by an appopriate citation (for example, we note that the statement at line 40 does not have a reference)

Additional Editor Comments (if provided):

- In the introduction and discussion, please include a new WHO recommendation on the use of masks in the community. How this revised recommendation may have impact on the effectiveness of a mask (Meff)?

- Please check the figure legends. Is there a typo in the description of Figure 2? Does Figure 1A refer to Figure 2A?

Reviewers' comments:

Reviewer's Responses to Questions

**Comments to the Author**

1. Is the manuscript technically sound, and do the data support the conclusions?

Reviewer #1: Yes

2. Has the statistical analysis been performed appropriately and rigorously? 

Reviewer #1: Yes

3. Have the authors made all data underlying the findings in their manuscript fully available?

Reviewer #1: Yes

4. Is the manuscript presented in an intelligible fashion and written in standard English?

Reviewer #1: No

5. Review Comments to the Author

Reviewer #1: The authors use a mathematical model to evaluate the impact of wearing masks on the burden of disease. Their model is well formulated, although rather simple. The results have good implications for public health policy and shows the effectiveness of wearing masks. I have some comments, and should these comments be addressed, I recommend the manuscript for publication.

1) Is it the case that the model uses a constant infectiousness rate throughout the natural course of infection? Does the level of infectivity change as the individual moves through their infection phase?

2) Building on the comment above, it has been established presymptomatics have a substantial role in disease transmission. The viral load is highest immediately before symptom onset. Since the authors used a SEIR model in which the exposed compartment is not transmitting, this is a major limitation. If feasible, the model should be extended to contain a presymptomatic compartment as well. If this is not feasible, the limitation should be discussed.

3) The predicted numbers seem too high and not very realistic. However, this is expected given the homogenous nature and random mixing of the population. This should be discussed.

4) There are a few typographical and grammar errors. For example, on line 121 the authors have "(COIVD, SRAS)".

6. PLOS authors have the option to publish the peer review history of their article (what does this mean?). If published, this will include your full peer review and any attached files.

Reviewer #1: No

---

## [Author Response · Author response to Decision Letter 0]

26 Jun 2020

Our responses are presented in our rebuttal letter.

---

## [Decision Letter · Decision Letter 1]

3 Aug 2020

Mask or No Mask for COVID-19: a Public Health and Market Study

PONE-D-20-11539R1

Dear Dr. Dai,

We’re pleased to inform you that your manuscript has been judged scientifically suitable for publication and will be formally accepted for publication once it meets all outstanding technical requirements.

Kind regards,

Kednapa Thavorn, PhD

Academic Editor

PLOS ONE

Additional Editor Comments (optional):

Reviewers' comments:

Reviewer's Responses to Questions

**Comments to the Author**

1. If the authors have adequately addressed your comments raised in a previous round of review and you feel that this manuscript is now acceptable for publication, you may indicate that here to bypass the “Comments to the Author” section, enter your conflict of interest statement in the “Confidential to Editor” section, and submit your "Accept" recommendation.

Reviewer #1: All comments have been addressed

2. Is the manuscript technically sound, and do the data support the conclusions?

Reviewer #1: Yes

3. Has the statistical analysis been performed appropriately and rigorously? 

Reviewer #1: Yes

4. Have the authors made all data underlying the findings in their manuscript fully available?

Reviewer #1: Yes

5. Is the manuscript presented in an intelligible fashion and written in standard English?

Reviewer #1: Yes

6. Review Comments to the Author

Reviewer #1: Dear authors,

Your comments are much appreciated. While the paper is simple in nature, it provides a useful narrative for the effectiveness of masks. Given the resurgence of the virus, not just in America but globally, the publication of this manuscript will further provide evidence of the effectiveness of masks in reducing transmission.

7. PLOS authors have the option to publish the peer review history of their article (what does this mean?). If published, this will include your full peer review and any attached files.

Reviewer #1: **Yes: **Affan Shoukat

---

## [Editor Report · Acceptance letter]

7 Aug 2020

PONE-D-20-11539R1 

Mask or No Mask for COVID-19: a Public Health and Market Study 

Dear Dr. Dai:

I'm pleased to inform you that your manuscript has been deemed suitable for publication in PLOS ONE. Congratulations! Your manuscript is now with our production department. 

Kind regards, 

on behalf of

Dr. Kednapa Thavorn 

Academic Editor

PLOS ONE